# ITA-IMMUNO-PET: The Role of [18F]FDG PET/CT for Assessing Response to Immunotherapy in Patients with Some Solid Tumors

**DOI:** 10.3390/cancers15030878

**Published:** 2023-01-31

**Authors:** Laura Evangelista, Andrea Bianchi, Alessio Annovazzi, Rosa Sciuto, Silvia Di Traglia, Matteo Bauckneht, Francesco Lanfranchi, Silvia Morbelli, Anna Giulia Nappi, Cristina Ferrari, Giuseppe Rubini, Stefano Panareo, Luca Urso, Mirco Bartolomei, Davide D’Arienzo, Tullio Valente, Virginia Rossetti, Paola Caroli, Federica Matteucci, Demetrio Aricò, Michelangelo Bombaci, Domenica Caponnetto, Francesco Bertagna, Domenico Albano, Francesco Dondi, Sara Gusella, Alessandro Spimpolo, Cinzia Carriere, Michele Balma, Ambra Buschiazzo, Rosj Gallicchio, Giovanni Storto, Livia Ruffini, Veronica Cervati, Roberta Eufrasia Ledda, Anna Rita Cervino, Lea Cuppari, Marta Burei, Giuseppe Trifirò, Elisabetta Brugola, Carolina Arianna Zanini, Alessandra Alessi, Valentina Fuoco, Ettore Seregni, Désirée Deandreis, Virginia Liberini, Antonino Maria Moreci, Salvatore Ialuna, Sabina Pulizzi, Maria Luisa De Rimini

**Affiliations:** 1Nuclear Medicine Unit, Department of Medicine DIMED, University of Padua, 35129 Padua, Italy; 2Nuclear Medicine Unit, ASO S.Croce e Carle Cuneo, 12100 Cuneo, Italy; 3Nuclear Medicine Unit, IRCCS Regina Elena National Cancer Institute, 00144 Rome, Italy; 4Department of Health Sciences (DISSAL), University of Genova, 16126 Genova, Italy; 5Nuclear Medicine Unit, IRCCS Ospedale Policlinico San Martino, 16132 Genoa, Italy; 6Section of Nuclear Medicine, Interdisciplinary Department of Medicine, University of Bari “Aldo Moro”, 70121 Bari, Italy; 7Nuclear Medicine Unit, Azienda Ospedaliero Universitaria di Modena, 41124 Modena, Italy; 8Nuclear Medicine Unit, University of Ferrara, 44121 Ferrara, Italy; 9Nuclear Medicine Unit, Dept Servizi Sanitari, AORN Ospedali dei Colli, 80131 Naples, Italy; 10Radiology Department, AORN Ospedali dei Colli, 80131 Naples, Italy; 11Nuclear Medicine Unit, IRCCS Istituto Romagnolo per lo Studio dei Tumori (IRST), 47014 Meldola, Italy; 12Nuclear Medicine Unit, Humanitas Istituto Clinico Catanese, 95045 Misterbianco, Italy; 13Nuclear Medicine Unit, University of Brescia, 25123 Brescia, Italy; 14Nuclear Medicine Department, Central Hospital Bolzano (SABES-ASDAA), 39100 Bolzano-Bozen, Italy; 15Dermatology Department, Central Hospital Bolzano (SABES-ASDAA), 39100 Bolzano-Bozen, Italy; 16Nuclear Medicine Unit, IRCCS CROB Referral Cancer Center of Basilicata, 85028 Rionero in Vulture, Italy; 17Nuclear Medicine Division, Azienda Ospedaliero-Universitaria of Parma, 43126 Parma, Italy; 18Department of Medicine and Surgery, Unit of Radiological Sciences, University of Parma, 43126 Parma, Italy; 19Nuclear Medicine Unit, Veneto Institute Of Oncology IOV—IRCSS, 35128 Padua, Italy; 20Nuclear Medicine Unit, ICS MAUGERI SPA SB—IRCCS, 35128 Padua, Italy; 21Nuclear Medicine Unit, Università degli Studi di Milano, Milano Statale, 20133 Milan, Italy; 22Nuclear Medicine Unit, Fondazione IRCCS Istituto Nazionale dei Tumori, 20133 Milan, Italy; 23Nuclear Medicine Division, Department of Medical Sciences, University of Turin, 10124 Turin, Italy; 24Nuclear Medicine Unit, Az. Ospedaliera Ospedali Riuniti Villa Sofia-Cervello di Palermo, 90100 Palermo, Italy

**Keywords:** immunotherapy, immune-related effects, therapy response, PET/CT, 18F-FDG

## Abstract

**Simple Summary:**

To examine the role of [18F]FDG PET/CT for assessing response to immunotherapy in patients with some solid tumors. Seventeen Italian centers analyzed the role of serial [18F]FDG PET/CT scans in patients candidates and, later undergoing immunotherapy for some solid cancers. Serial [18F]FDG PET/CT can be useful in evaluating the response to therapy, soon after 3 and 6-months from the start of immunotherapy. The evidences were foud both in patients affected by lung cancer and malignant melanoma, although large prospective trials are needed for definitively confirmed these findings.

**Abstract:**

AIM: To examine the role of [18F]FDG PET/CT for assessing response to immunotherapy in patients with some solid tumors. METHODS: Data recorded in a multicenter (*n* = 17), retrospective database between March and November 2021 were analyzed. The sample included patients with a confirmed diagnosis of a solid tumor who underwent serial [18F]FDG PET/CT (before and after one or more cycles of immunotherapy), who were >18 years of age, and had a follow-up of at least 12 months after their first PET/CT scan. Patients enrolled in clinical trials or without a confirmed diagnosis of cancer were excluded. The authors classified cases as having a complete or partial metabolic response to immunotherapy, or stable or progressive metabolic disease, based on a visual and semiquantitative analysis according to the EORTC criteria. Clinical response to immunotherapy was assessed at much the same time points as the serial PET scans, and both the obtained responses were compared. RESULTS: The study concerned 311 patients (median age: 67; range: 31–89 years) in all. The most common neoplasm was lung cancer (56.9%), followed by malignant melanoma (32.5%). Nivolumab was administered in 46.3%, and pembrolizumab in 40.5% of patients. Baseline PET and a first PET scan performed at a median 3 months after starting immunotherapy were available for all 311 patients, while subsequent PET scans were obtained after a median 6, 12, 16, and 21 months for 199 (64%), 102 (33%), 46 (15%), and 23 (7%) patients, respectively. Clinical response to therapy was recorded at around the same time points after starting immunotherapy for 252 (81%), 173 (56%), 85 (27%), 40 (13%), and 22 (7%) patients, respectively. After a median 18 (1–137) months, 113 (36.3%) patients had died. On Kaplan–Meier analysis, metabolic responders on the first two serial PET scans showed a better prognosis than non-responders, while clinical response became prognostically informative from the second assessment after starting immunotherapy onwards. CONCLUSIONS: [18F]FDG PET/CT could have a role in the assessment of response to immunotherapy in patients with some solid tumors. It can provide prognostic information and thus contribute to a patient’s appropriate treatment. Prospective randomized controlled trials are mandatory.

## 1. Introduction

The introduction of immunotherapy in the fight against tumors has had some beneficial effects on the outcome of various solid tumors. Starting with the approval of anti-cytotoxic T lymphocyte-associated protein 4 (anti-CTLA-4) for advanced metastatic malignant melanoma in 2011, immune checkpoint inhibitors (ICIs) have since included antibodies against programmed cell death 1 (PD-1) as well. Its ligand (PDL-1) quickly gained US Food and Drug Administration approval for the treatment of a wide array of cancer types, demonstrating a strong impact on patient survival [1,2,3,4,5,6,7,8].

Which patients can benefit from a durable response to ICIs is still a clinical issue, however. The onset of immune-related adverse events can also complicate the duration and efficacy of such treatments [9]. Response to immunotherapy is currently judged on the basis of clinical assessments supported by imaging findings, mainly using contrast-enhanced computed tomography (CT) and standardized criteria (i.e., iRECIST, iRC; [10]). The complexity of the diverse responses to immunotherapy makes using conventional morphological criteria a challenge, however, and several studies have used [18F]FDG PET/CT in efforts to monitor and predict response to ICIs [11,12,13,14,15,16,17,18,19]. Despite numerous reports relating to this endpoint, a study on a large population undergoing serial [18F]FDG PET/CT before and while receiving ICIs is still lacking, and some clinical questions remain unanswered.

The primary objective of the present study was therefore to examine the role of [18F]FDG PET/CT for assessing response to immunotherapy in a group of patients with some solid cancer tumors. As additional endpoints, we aimed to: (1) correlate prognosis with [18F]FDG PET/CT findings in patients given immunotherapy; and (2) identify the ability of [18F]FDG PET/CT to detect adverse events associated with immunotherapy.

## 2. Materials and Methods

### 2.1. Study Approval and Patient Population

The study protocol was approved by the institutional ethical committee at the “dei Colli” Hospital (Napoli, Italy) in July 2020 (No. AOC-0020062-2020). Seventeen Italian centers with a well-established experience in PET/CT and oncology took part in the study. A dedicated Microsoft Excel datasheet was created for the purposes of data collection. Any anomalies identified were clarified by interviewing the centers involved before conducting the analysis.

### 2.2. Patient Population

We retrospectively reviewed [18F]FDG PET/CT scans obtained for 374 consecutive patients with various solid tumors at the Nuclear Medicine Units of the 17 participating centers (Appendix A) enrolled from August, 2013 to December, 2020. 

The following inclusion criteria were adopted: (1) a confirmed diagnosis of tumor in patients who underwent serial [18F]FDG PET/CT before and after one or more cycles of immunotherapy; (2) age >18 years; and (3) a follow-up of at least 12 months after the first PET/CT scan. Patients enrolled in clinical trials or without a confirmed diagnosis of cancer were excluded.

### 2.3. PET/CT Equipment and Image Acquisition Protocol

Similar standard protocols, all in compliance with EANM procedural guidelines [20], were used at all centers for PET/CT image acquisition. All patients fasted for at least 6 hours prior to imaging, and their blood glucose levels were <200 mg/dL at the time of tracer injection. To minimize [18F]FDG uptake in skeletal muscle, all patients were instructed to avoid talking, chewing, or any muscular activity before acquiring the PET/CT scan. The PET/CT studies were acquired and integrated using PET/CT systems, according to the protocols adopted at each participating center. PET data on whole-body tracer distribution were then acquired (3 min per bed) in 3-D mode, starting 60 min after the iv. administration of [18F]FDG. Attenuation was corrected using CT images. CT and PET images were matched and fused into transaxial, coronal, and sagittal images. All images were analyzed by two experienced nuclear medicine specialists for each of the 17 centers, who were unaware about the clinical data.

### 2.4. Interpretation of PET/CT Images 

Serial [18F]FDG PET/CT scans were performed after starting immunotherapy. Data were collected from PET scans performed up until the fifth time after starting immunotherapy. The scans were all named as follows: baseline PET (before starting immunotherapy); PET1 (at a median 3 months, range 2–4, after starting immunotherapy); PET2 (at a median 6 months, range 5–8); PET3 (at a median 12 months, range 9–14); PET4 (at a median 16 months, range: 14–18); and PET5 (at a median 21 months, range 19–24). At each time point after starting immunotherapy, cases were classified as: complete metabolic response (CMR), partial metabolic response (PMR), stable metabolic disease (SMD), or progressive metabolic disease (PMD), based on visual and semiquantitative analyses, according to the EORTC criteria [21,22]. For the semiquantitative analysis, SUVmax was obtained from the target lesions at baseline and on every subsequent follow-up scan. No specific cut-off value for SUVmax was used in each PET scan. “Treatment metabolic response or metabolic responders” was defined as CMR + PMR, while “metabolic disease control” was considered as a combination of CMR + PMR + SMD. Conversely, PMD + SMD and PMD alone were defined, respectively, as “no-metabolic treatment response or non-responders” and “non metabolic disease control”. 

There is currently no standard definition for immunotherapy-induced organ inflammation on [18F]FDG PET/CT, therefore our definition was based on the visual finding of a diffuse and homogeneous increase in the intensity of tracer uptake in a given organ by comparison with the baseline PET/CT scan. Tracer uptake was recorded separately for each organ (as thyroiditis, colitis, pneumonitis, etc.).

### 2.5. Clinical Response to Immunotherapy

Patients were generally monitored according to current clinical guidelines. The information obtained at the clinical assessments included: impressions on physical examination; routine blood work and serum chemistry studies; conventional imaging findings at the medical oncologists’ discretion. Clinical assessments were recorded in terms of stable disease, improving or worsening symptoms, or biochemical data. The definition of “clinical response” was made in case of improvement, while a “clinical disease control” was considered in case of stable and improved health condition.

### 2.6. Follow-Up

The follow-up data for our selected sample of patients were obtained from clinical charts or by means of telephone interviews. The date of the latest clinical examination or consultation was used to establish the length of follow-up. Overall survival (OS) was defined as the time elapsing between baseline [18F]FDG PET/CT scans and all-cause mortality. Deaths were certified. 

### 2.7. Statistical Analysis

Appendix A reported the study design. Categorical variables are reported as frequencies, and continuous variables as means with the standard deviation (SD) for variables with a normal distribution, and as medians and interquartile ranges (IQRs) for variables with a non-normal distribution. Student’s *t*-test was used to compare continuous variables, where appropriate. Pearson’s chi-square test was used to compare categorical variables. ANOVA test with the Bonferroni correction was used for comparing more than 2 groups. The agreements between categorial variables were tested by using K-statistics (Cohen index). Values ≤ 0 indicated no agreement, 0.01–0.20 as none to slight, 0.21–0.40 as fair, 0.41–0.60 as moderate, 0.61–0.80 as substantial, and 0.81–1.00 as almost perfect agreement. OS curves for each category were computed using Kaplan–Meier analysis, and the log-rank test was used to compare them. Cox’s regression analysis was used to assess predictors of OS. The statistical analyses were performed using SPSS version 19 (IBM, Armonk, NY, USA), and MedCalc version 20.027 (MedCalc Software Ltd, Ostend, Belgium).

## 3. Results

### 3.1. Patient Population

After applying our inclusion and exclusion criteria, 311 patients were analyzed for the endpoints of the present study. Figure 1 shows the flowchart for the selection of the final study population. Details of the study population are shown in Table 1. 

Most patients had lung cancer (*n* = 177; 56.9%) or metastatic malignant melanoma (*n* = 101, 32.5%). Most of them had been treated before starting immunotherapy (*n* = 269, 86.5%), most often with nivolumab or pembrolizumab (46.3% of lung cancer patients, and 40.5% of malignant melanoma patients). Immunotherapy was used alone in 81.4% of cases and administered concomitantly with other treatments in the other 18.6%. Based on the type of malignancy, some differences emerged, mainly as regards to the type of immunotherapy, previous treatments, and the presence of comorbidities. All 311 patients underwent baseline PET/CT (time between baseline PET/CT and the start of immunotherapy ranged between 1 and 2 months), and they all had a PET1 scan. The numbers of patients having further PET scans decreased over time as follows: 199 (64%) had a PET2; 102 (33%) had a PET3; 46 (15%) had a PET4; and 23 (7%) had a PET5. 

### 3.2. Clinical and PET/CT Response to Immunotherapy

Clinical responses to therapy were available for 252 (81%), 173 (56%), 85 (27%), 40 (13%), and 22 (7%) patients, respectively, at the first, second, third, fourth, and fifth assessments after starting immunotherapy. The number of patients in the serial PET and clinical assessment was reduced also in accordance with the progression of disease (rate of clinical progression: 16%, 21%, 11%, 10%, and 9% at a median 3, 6, 12, 16, and 21 months after starting immunotherapy, respectively). Table 2 shows the rates of response and disease control to immunotherapy based on [18F]FDG PET/CT findings and clinical assessments at each time point, and for all patients and based on the different cancer type. 

One-hundred twenty-two patients were classified as having PMD on their PET1 scans, even though their oncologists reported a clinical improvement. There were more cases of SMD, judging from both PET scans and clinical findings, at the time of PET2, PET3, and PET4. In terms of disease control, the consistency (K value) between the first, second, third, fourth, and fifth PET scans and the clinical assessments, at each time point was 0.34, 0.54, 0.26, 0.37, and 0.33, respectively (Appendix A). Conversely in terms of response to therapy, K values were 0.27, 0.25, 0.29, and 0.14, respectively, at each time.

When patients were analyzed separately by type of tumor, most of the patients with PMD at PET1 had malignant melanoma, whereas patients with lung cancer or other solid tumors achieved higher PMR rates (35% and 45.5%, respectively). Metabolic responses shifted more towards PMD after the second and third PET scans. Interestingly, in patients with lung, genito-urinary, head and neck, breast, and gastrointestinal cancer, [18]FDG uptake in the index lesions at baseline PET, at PET1, and PET 2 was slightly higher than patients with malignant melanoma, although not significantly different. 

Forty-six (38%) of 122 patients with PMD on their PET1 scans discontinued the immunotherapy after their first assessment (25 patients with lung cancer, 13 with malignant melanoma, and eight with other solid tumors). PMD on PET2 scans was likewise associated with a subsequent discontinuation of immunotherapy for 30/82 (37%) patients (20 patients with lung cancer, six with malignant melanoma, and four with other solid tumors). The decision to stop immunotherapy was also associated with worsening clinical conditions in 56% and 45% of patients at their first and second assessments, respectively. We noted, however, that 68% (*n* = 45/66) of patients with signs of PMD on their PET1 scan had confirmed PMD on their PET2 scan (Appendix A). Conversely, 11/66 (17%) patients showed a CMR or PMR at PET2, therefore denoting a potential pseudoprogression at [18]F-FDG PET/CT scan. Four out of 11 (36%) patients were affected by lung cancer, while 7/11 (64%) by malignant melanoma. In this latter group of patients, all underwent anti-CTLA4 therapy. The false interpretation of pseudoprogression was relative to the appearance of a slight uptake of [18]F-FDG in some lymph nodes or in small lung nodules, respectively, for patients affected by malignant melanoma and lung cancer.

### 3.3. Follow-Up

At a median follow-up of 18 months (1–137 months), 113 patients (36.3% of the total) had died: 74/177 with lung cancer, 33/101 with malignant melanoma, and 6/33 with other solid tumors. Data were missing for 35 patients (11.3%). The median survival time was significantly longer for patients showing a metabolic disease control on their PET1 and PET2 scans (Appendix A). On the other hand, the median survival time slightly differed between patients with versus without a disease control at the first and second assessment after starting immunotherapy (Appendix A), which was similarly reported for responders and non-responders patients at PET1 and PET2 and after the first and the second clinical evaluations.

As illustrated, three months after starting immunotherapy, Kaplan–Meier analysis demonstrated significant differences in survival between patients with PET scans showing signs of CMR, PMR, SMD, or PMD. On the other hand, although significant, an overlap among patients who had a clinical stable, improved, or worsened condition was demonstrated (Figure 2A; log rank test: 59.31, *p* < 0.0001 vs. 9.92, *p* < 0.05, respectively, for PET and clinical assessment). 

This situation was more obvious for patients with lung cancer than for those with malignant melanoma or other solid tumors (Figure 3). 

Six months after starting immunotherapy, however, clinical findings and PET results were both significant in stratifying survival rates for all patients (Figure 2B, log rank test: 39.20, *p* < 0.0001 vs. 57.56, *p* < 0.0001, respectively, for PET and clinical assessment). Composed Kaplan–Meier analyses were performed by matching clinical and PET data after a median of 3 months from the start of immunotherapy, as illustrated in Appendix A. The OS was better in patients with a response or a disease control both at [18F]FDG PET/CT and clinical assessment; interestingly, patients with a clinical response or a clinical disease control but no metabolic response or no metabolic disease control at PET showed a worse prognosis. This latter result underlined the complementary role of metabolic imaging in the first 3 months from the start of immunotherapy.

Finally, at univariate analysis, the following were significant predictors of OS: immunotherapy in combination with other treatments; other treatments prior to immunotherapy; and response to immunotherapy on PET1, PET2, and PET3 scans, and at the corresponding clinical assessments (Appendix A). None of these variables emerged as independent predictors of survival at multivariate analysis, however. 

### 3.4. Inflammatory Side Effects of Immunotherapy and [18F]FDG PET/CT

Signs of inflammation were reported on the PET1 scans of 54/311 (17%) patients, 37/177 with lung cancer, 14/101 with malignant melanoma, and 3/33 with other solid tumors. These were mainly cases of pulmonitis (46.2%) or thyroiditis (22.2%), followed by colitis and other types of inflammation (i.e., lymphadenitis, osteoarticular inflammation, esophagitis). The inflammatory condition was only confirmed in 24/54 patients (44.4%). Inflammation was detected on PET2 scans in 31/199 (15.6%) patients, 22/101 with lung cancer, 7/77 with malignant melanoma, and 2/21 with other solid tumors. Here again, pulmonitis was the most common inflammatory side effect (12/31), followed by the above-mentioned other types (*n* = 9; 29%), colitis (*n* = 6; 19%), and thyroiditis (*n* = 4; 13%). The inflammation seen on PET2 scans was confirmed by clinical/imaging findings in 19/31 (61.3%) patients. 

The OS curve was significantly better for patients with no sign of inflammation on their PET1 scan (*p* = 0.032), but this statistical difference was lost in cases with confirmed and unconfirmed clinical/imaging inflammation. 

## 4. Discussion

The present study concerns a large population of patients with some solid tumors who underwent serial [18F]FDG PET/CT scans before and during immunotherapy. We demonstrated that assessing response to therapy on PET scans according to EORTC criteria can stratify patient outcomes better than clinical findings, especially after 3 months from the start of immunotherapy. [18F]FDG PET/CT can therefore facilitate an early distinction between patients who will or will not benefit from immunotherapy. However, these considerations should be confirmed by prospective clinical trials.

In these times of evidence-based medicine, it is crucial for healthcare professionals to be able to rely on medical guidelines and standardized criteria to improve the quality and consistency of patient care.Much effort has gone into standardizing the assessment of response to immunotherapy [23,24,25], and new criteria have been tested in patients with metastatic melanoma or lung cancer. Currently, standard of care for response assessment is clinical assessment and conventional imaging (i.e., CT by using modified RECIST criteria), however, in the present study, we aimed to compare the clinical assessment with metabolic evaluation rather than the morphological one.

Cho et al. [11] focused on the sensitivity and specificity of such new criteria and found the PET/CT criteria for the early prediction of response to immune checkpoint inhibitor therapy (PECRIT) more sensitive, specific, and accurate than EORTC criteria (100%, 93%, and 95% vs. 40%, 73%, and 65%), when applied to metastatic melanoma patients. In the same disease setting, Sachpedikis et al. [26] likewise compared the efficacy of EORTC criteria with another approach, PET Response Evaluation Criteria for ImmunoTherapy (PERCIMT), applied to early [18F]FDG PET/CT images. PERCIMT proved significantly more sensitive than EORTC (93.6% vs. 64.5%) but showed a similar specificity (70% vs. 90%). Neither Cho et al. [11] nor Sachpedikis et al. [26] included a long-term follow-up to test these new criteria for the purposes of predicting outcome, but they did compare metabolic findings with clinical benefit. A recent meta-analysis confirmed that the modified criteria exhibited a better pooled sensitivity (94% vs. 64%) and specificity (84% vs. 80%) than conventional EORTC criteria in patients with metastatic melanoma [27], although no information was reported on long-term outcomes. Goldfarb et al. [28] developed the iPERCIST criteria (a combination of the iRECIST and PERCIST criteria) for patients with lung cancer. The authors found a longer survival time in metabolic responders than in non-responders (94% vs. 11%, respectively) at 1-year follow-up. There is still not enough data to show conclusively which of these proposed criteria are superior. Some studies have compared different criteria [8], but large prospective trials will be needed to validate them. The impact on long-term patient outcomes has yet to be prospectively validated in randomized clinical trials [29]. The applicability of any new criteria in clinical practice should be addressed as well because the process is time-consuming. PERCIST criteria are also sometimes difficult to apply without dedicated systems. The recently published joint EANM/SNMMI/ANZSNM practice guidelines on the use of [18F]FDG PET/CT imaging during immunomodulatory treatments made the point that the effects of such therapy on cancer patients are often assessed in the context of busy PET centers, so they should be user-friendly, and based on reliable PET metrics. For now, no validated model has become available for use in this dynamic research field [29]. Herein, we found that, although less sensitive than the other criteria, EORCT criteria were able to stratify the mortality risk in a population of more than 300 patients undergoing immunotherapy. 

One of the most important challenges for assessing response to immunotherapy concerns the presence of pseudoprogression after the first treatment cycles. Most studies testing the new criteria have overlooked this issue. In the present experience, it emerged that 11/66 patients with PMD at PET1 had a CMR/PMR at PET2, meaning that about 17% of patients showed a pseudoprogression between 3 and 6 months after starting immunotherapy, in line with the current available data [30]. Pseudoprogression has been mainly reported in melanoma patients receiving anti-CLTA4 agents, with approximately 15% of patients experiencing pseudoprogression [31]. Pseudoprogression appears to be much rarer in all other tumor types (less than 3%), especially with the use of anti-PD1/PD-L1 agents, indicating that in most of the patient’s progression seen on morphological imaging is authentic progression [32]. Indeed, we found that pseudoprogression was more frequent in malignant melanoma and mainly in those patients treated with anti-CTLA4 agents. Therefore, a careful analysis of PET images is essential for the correct interpretation; it is important to suggest a further metabolic evaluation in case of findings suspected for pseudopregression. In our clinical experience, the appearance of a slight [18]F-FDG uptake in small lymph nodes close to the primary malignant melanoma lesion or close to the other metastatic nodes or in small lung nodules or in pleaural thickening at PET1 scan was often suggestive of pseudoprogression. 

The complementary role of clinical and imaging findings can be important in this setting, as only 38% of our patients with PMD at PET1 discontinued the treatment. Three months after starting immunotherapy, PET imaging can better stratify OS, especially in patients with lung cancer (see Figure 3). Furthermore, the identification of a specific subgroup of patients, i.e., those with a clinical response but no response or no metabolic disease control at PET after 3 months from the start of immunotherapy (Appendix A) could be helpful to drawn a personalized therapeutical approach.

There were signs of inflammation on the first [18F]FDG PET/CT scan in 17% of all the entire population, but they were confirmed by clinical/imaging findings in 44.4% of them. On the second PET scan, the proportion of patients with evidence of inflammation decreased to 15.6%, but it was confirmed in 61.3% of cases. Immune-related side effects reportedly occur more frequently with anti-CTLA4 than with anti-PD-1/PD-L1 therapies [33]. In the present study, only 20 patients underwent anti-CTLA4, while 291 were treated with an anti-PD-1/PD-L1 therapy. The rates of adverse events at PET/CT were 10% and 17.5%, respectively, for patients treating with anti-CTLA4 and anti-PD-1/PD-L1 drugs. Due to the unabalanced number of enrolled patients, we cannot add any additional information. However, adverse events most commonly occur within 3–6 months after starting therapy [34], and usually resolve within 12 weeks after the onset of symptoms [35]. In the present study too, immune-related side effects came to light during the 3–6 months after starting of systemic therapy (between PET1 and PET2). This evidence should be translated into clinical practice: at the time of PET2, any findings suggestive of adverse effects of the immunotherapy should be carefully assessed and confirmed. Although no associations have been reported between PET-related colitis or diarrhea and response to therapy [36], [18F]FDG PET/CT might precede a clinical diagnosis, as already suggested by Wong et al. [37]. The correlation between the development of adverse events to immunotherapy and the response to therapy is controversial in literature; some authors reported no correlation [36], some authors found a significant correlation [27]. In the present study, the evidence of metabolic signs suggestive for adverse events at the first PET scan after the start of immunotherapy was significantly correlated with a better OS, but the significance was lost after confirming or not PET positivity. In a prospective trial, this aspect should be considered and solved.

The present study has some limitations. The first concerns the retrospective design; the second the heterogeneity in patients’ treatments before or during the administration of immunotherapy. Third, there is the issue of the small number of patients with diseases other than lung cancer of malignant melanoma, and the diverse biological characteristics of the other solid tumors considered. Fourth, the response to criteria that was used in the present study. Indeed, EORTC response assessment only considers changes in the SUVmax of the target lesions with no consideration of the change in size on morphologic imaging. Probably, the relatively high rate of pseudoprogression in this study (17%) would be reduced by using other response criteria such as PERCIST. Nevertheless, as already mentioned in the discussion part, PERCIST criteria are difficult to apply without dedicated systems, thus being time consuming and very difficult to be used in daily clinical practice. Finally, the absence of the central evaluation of the images. However, the data were collected by centers and personnel with an experience in oncological [18F]FDG PET/CT imaging of more than 10 years.

## 5. Conclusions

[18F]FDG PET/CT could have a role in assessing response to immunotherapy in patients with solid tumors, particularly in cases of lung cancer. Already at 3 months after starting immunotherapy, it can provide prognostic information and thus contribute to a patient’s most appropriate therapeutic management. Due to delay of action for immunotherapy for a period of time between 2 to 6 months, additional PET parameters should be defined in order to find a more appropriate biomarker for the oncological point of view. Therefore, prospective randomized controlled trials are needed to establish whether assessing response to therapy early on a combination of [18F]FDG PET/CT and clinical indicators can affect the further management of patients given immunotherapy, and whether it can change their prognosis. 

## Figures and Tables

**Figure 1 cancers-15-00878-f001:**
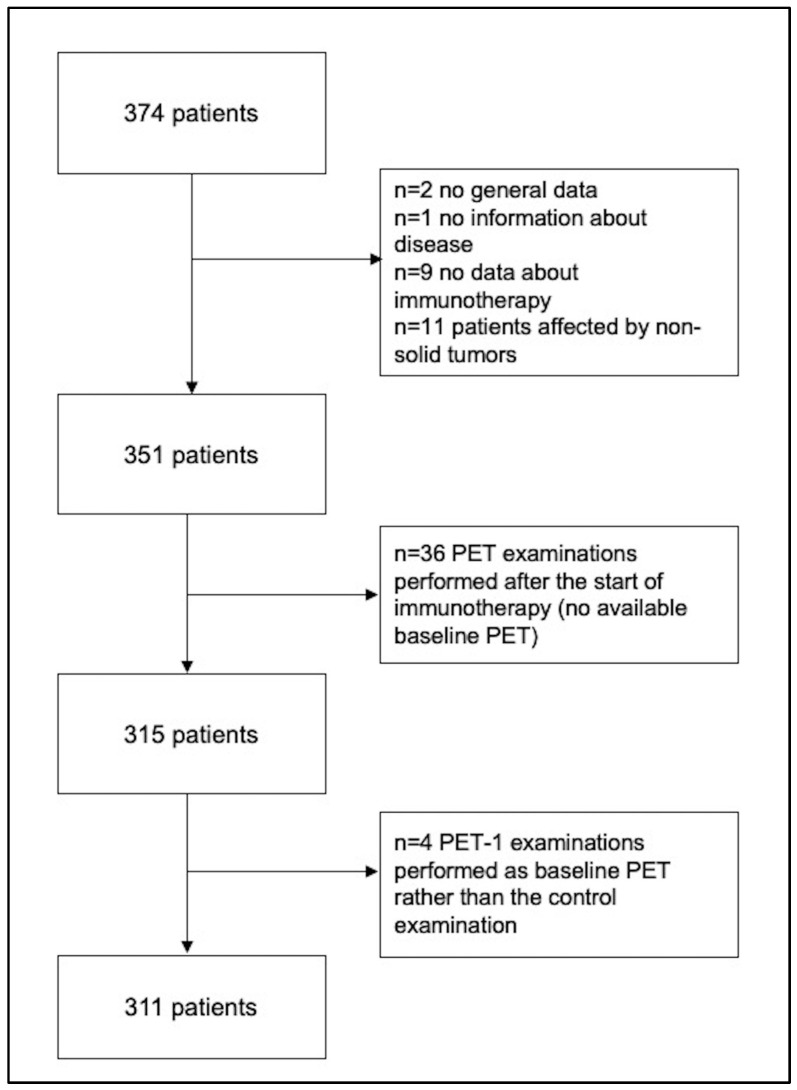
Flowchart for the selection of the final population.

**Figure 2 cancers-15-00878-f002:**
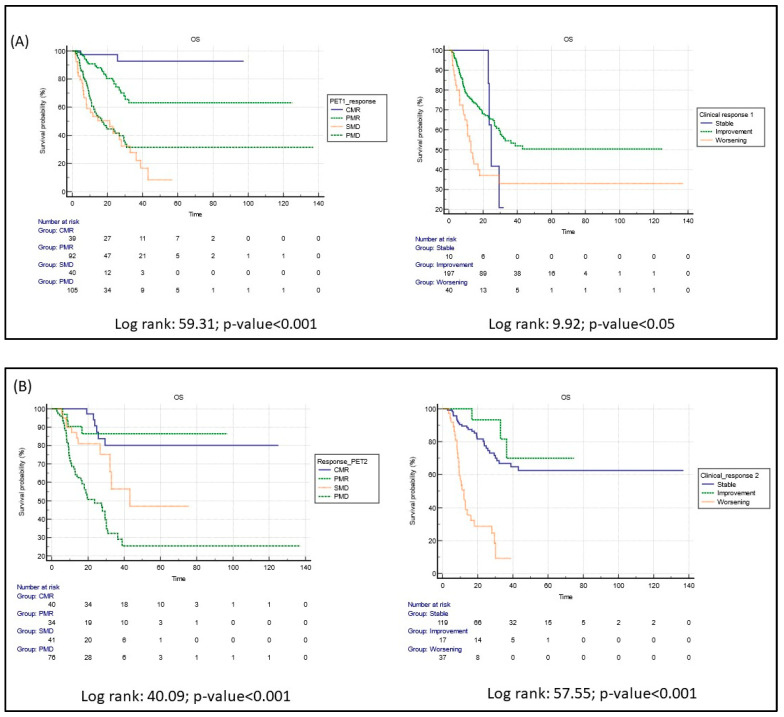
(**A**) Kaplan–Meier curves for OS based on the first PET/CT scan and clinical examination after starting immunotherapy in 276 patients; (**B**) Kaplan–Meier curves for OS based on the second PET/CT scan and clinical examination in 190 patients. CMR = complete metabolic response; PMR = partial metabolic response; SMD = stable metabolic disease; PMD = progressive metabolic disease; Time was expressed in months.

**Figure 3 cancers-15-00878-f003:**
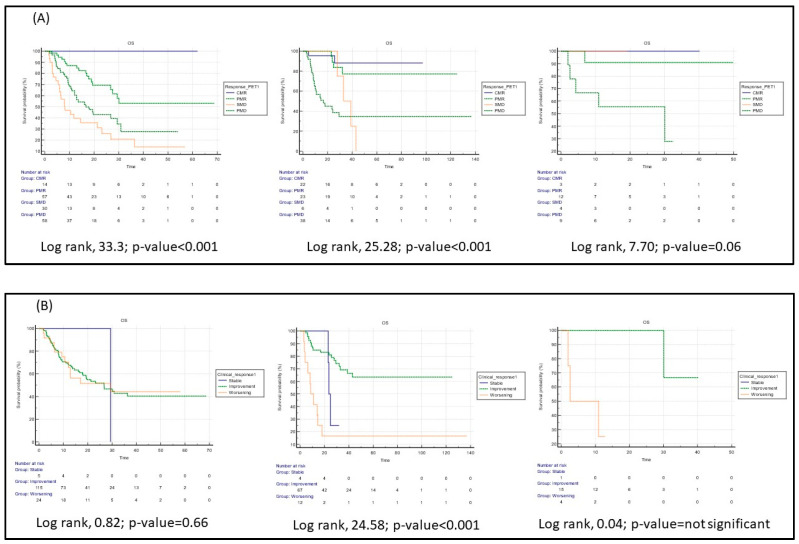
Kaplan–Meier curves for OS based on the first PET/CT scan (**A**) and clinical examination (**B**), by type of cancer (*n* = 159 patients with lung cancer, *n* = 89 patients with malignant melanoma and *n* = 28 patients with other tumors). CMR = complete metabolic response; PMR = partial metabolic response; SMD = stable metabolic disease; PMD = progressive metabolic disease; Time was expressed in months.

**Table 1 cancers-15-00878-t001:** Characteristics of patient population.

Variables	All Patients	Lung Cancer	Malignant Melanoma	Others
*n*	311	177	101	33
Median age (range), years	67 (31–89)	68 (44–86)	67 (31–89)	67 (35–84)
Type of disease, *n* (%)				
Lung cancer	177 (56.9%)	177 (100%)	-	-
Melanoma	101 (32.5%)	-	101 (100%)	-
Other solid cancers				
Genito-urinary	11 (3.5%)	-	-	11 (33.3%)
Head and neck	13 (4.2%)	-	-	13 (39.3%)
Breast cancer	3 (1%)	-	-	3 (9.2%)
Gastrointestinal tract	6 (2%)	-	-	6 (18.2%)
Comorbidity, *n* (%)				
No	71 (22.8%)	59 (33%)	4 (4%)	8 (24%)
Yes	135 (43.4%)	84 (48%)	34 (34%)	17 (52%)
Not available	105 (33.8%)	34 (19%)	63 (63%)	8 (24%)
Treatments before immunotherapy, *n* (%)				
No	18 (5.8%)	18 (10.2%)	0	0
Yes				
Surgery	180 (57.9%)	54 (30.5%)	101 (100%)	25 (75.8%)
RT	46 (14.8%)	32 (18.1%)	1 (1%)	13 (39.4%)
Chemotherapy	162 (52.1%)	135 (76.3%)	2 (2%)	26 (78.8%)
Combination of local and systematic therapies	94 (30.2%)	67 (37.9%)	3 (3%)	24 (72.7%)
Not available	9 (2.9%)	9 (5.1%)	0	0
Type of immunotherapy, *n* (%)				
Atezolizumab	18 (5.8%)	14 (7.9%)	0	4 (12.1%)
Nivolumab	144 (46.3%)	82 (46.3%)	42 (41.6%)	20 (60.6%)
Durvalumab	1 (0.3%)	1 (0.6%)	0	0
Ipilimumab	20 (6.4%)	0	20 (19.8%)	0
Pembrolizumab	126 (40.5%)	79 (44.6%)	39 (38.6%)	8 (24.2%)
Cemiplimab	2 (0.6%)	1 (0.6%)	0	1 (3%)
Rate of immunotherapy administration, *n* (%)				
Weekly	2 (0.6%)	2 (1%)	0	0
Two-weekly	113 (36.3%)	76 (42%)	19 (19%)	18 (55%)
Three-weekly	140 (45%)	86 (49%)	49 (49%)	5 (15%)
Others	40 (12.9%)	10 (6%)	23 (23%)	7 (21%)
Not available	16 (5.1%)	3 (2%)	10 (10%)	3 (9%)
Combination of immunotherapy and other treatments, *n* (%)				
No	253 (81.4%)	39 (22%)	13 (12.9%)	6 (18.2%)
Yes	58 (18.6%)	138 (78%)	88 (87.1%)	27 (81.8%)

**Table 2 cancers-15-00878-t002:** PET and clinical responses.

Variables	All Patients	Lung Cancer	Malignant Melanoma	Others
	(*n* = 311)	(*n* = 177)	(*n* = 101)	(*n* = 33)
PET response 1, *n* (%)				
CMR	44 (14.1%)	17 (9.6%)	24 (23.8%)	3 (9.1%)
PMR	101 (32.5%)	62 (35%)	24 (23.8%)	15 (45.5%)
SMD	44 (14.1%)	33 (18.6%)	7 (6.9%)	4 (12.1%)
PMD	122 (39.2%)	65 (36.7%)	46 (45.5%)	11 (33.3%)
Treatment response at PET response 1, *n* (%)				
Responders (CMR, PMR)	145 (46.6%)	79 (44.6%)	48 (47.5%)	18 (54.5%)
No responders (SMD, PMD)	166 (53.4%)	98 (55.4%)	53 (52.5%)	15 (45.5%)
Disease control at PET response 1, *n* (%)				
Disease control	189 (60.8%)	112 (63.3%)	55 (54.5%)	22 (66.7%)
(CMR, PMR, SMD)				
No disease control (PMD)	122 (39.2%)	65 (36.7%)	46 (45.5%)	11 (33.3%)
Clinical Response 1, *n* (%)				
Stable disease	10 (3.2%)	5 (3%)	4 (4%)	1 (3%)
Clinical improvement	202 (65%)	119 (67%)	67 (66%)	16 (48%)
Clinical worsening	40 (12.9%)	24 (14%)	12 (12%)	4 (12%)
Not available	59 (19%)	29 (16%)	18 (18%)	12 (36%)
Clinical response 1, *n* (%)				
Responders	202 (65%)	119 (67%)	67 (66%)	16 (49%)
(improvement)				
No Responders	50 (16%)	29 (17%)	16 (16%)	5 (15%)
(stable + worsening)				
Not available	59 (19%)	29 (16%)	18 (18%)	12 (36%)
Disease control at Clinical 1, *n* (%)				
Disease control	212 (68.2%)	124 (70%)	71 (70%)	17 (52%)
(stable + improvement)				
No disease control	40 (12.9%)	24 (14%)	12 (12%)	4 (12%)
(worsening)				
Not available	59 (19%)	29 (16%)	18 (18%)	12 (36%)
PET response 2, *n* (%)				
CMR	41 (13.2%)	13 (7%)	26 (26%)	2 (6%)
PMR	34 (10.9%)	18 (10%)	12 (12%)	4 (12%)
SMD	42 (13.5%)	26 (15%)	12 (12%)	4 (12%)
PMD	82 (26.4%)	44 (25%)	27 (27%)	11 (33%)
Not available	112 (36%)	76 (42%)	35 (35%)	12 (36%)
Treatment response at PET response 2, *n* (%)				
Responders (CMR, PMR)	75 (24.1%)	31 (17.5%)	38 (37.6%)	6 (18.2%)
No responders (SMD, PMD)	124 (39.9%)	70 (39.5%)	39 (38.6%)	15 (45.5%)
Not available	112 (36%)	76 (42%)	24 (24%)	12 (36.4%)
Disease control at PET response 2, *n* (%)				
Disease control	117 (37.6%)	57 (32%)	50 (50%)	10 (30%)
(CMR, PMR, SMD)				
No disease control (PMD)	82 (26.4%)	44 (25%)	27 (27%)	11 (33%)
Not available	112 (36%)	76 (42%)	24 (24%)	12 (36%)
Clinical Response 2, *n* (%)				
Stable disease	119 (38.3%)	55 (31%)	50 (50%)	14 (42%)
Clinical improvement	17 (5.5%)	10 (6%)	7 (7%)	0
Clinical worsening	37 (11.9%)	21 (12%)	14 (14%)	2 (6%)
Not available	138 (44.4%)	91 (51%)	24 (24%)	17 (52%)
Clinical response 2, *n* (%)				
Responders (improvement)	17 (5.5%)	10 (6%)	7 (7%)	0
No Responders	136 (50.1%)	76 (43%)	64 (69%)	16 (48%)
(stable + worsening)				
Not available	138 (44.4%)	91 (51%)	24 (24%)	17 (52%)
Clinical disease control 2 (categorical data), *n* (%)				
Disease control	136 (43.7%)	65 (37%)	57 (57%)	14 (42%)
(Stable, improvement)				
No disease control	37 (11.9%)	21 (12%)	14 (14%)	2 (6%)
(worsening)				
Not available	138 (44.4%)	91 (51%)	30 (30%)	17 (52%)
PET response 3, *n* (%)				
CMR	25 (8%)	4 (2%)	17 (17%)	4 (12%)
PMR	8 (2.6%)	4 (2%)	2 (2%)	2 (6%)
SMD	35 (11.3%)	19 (11%)	12 (12%)	4 (12%)
PMD	34 (10.9%)	17 (10%)	14 (14%)	3 (9%)
Not available	209 (67.2%)	133 (75%)	56 (55%)	20 (61%)
Treatment response at PET response 3, *n* (%)				
Responders (CMR, PMR)	33 (10.6%)	8 (4.5%)	19 (18.8%)	6 (18.2%)]
No responders (SMD, PMD)	69 (22.2%)	15 (8.5%)	26 (25.7%)	7 (21.2%)
Not available	209 (67.2%)	133 (75%)	56 (55.4%)	20 (60.6%)
Disease control at PET response 3, *n* (%)				
Disease control	68 (21.9%)	27 (15%)	31 (31%)	10 (30%)
(CMR, PMR, SMD)				
No disease control (PMD)	34 (10.9%)	17 (10%)	14 (14%)	3 (9%)
Not available	209 (67.2%)	133 (75%)	56 (55%)	20 (61%)
Clinical Response 3, *n* (%)				
Stable disease	69 (22.2%)	26 (15%)	33 (33%)	10 (30%)
Clinical improvement	7 (2.3%)	3 (2%)	3 (3%)	1 (3%)
Clinical worsening	9 (2.9%)	4 (2%)	4 (4%)	1 (3%)
Not available	226 (72.7%)	144 (81%)	61 (61%)	21 (64%)
Clinical response 3, *n* (%)				
Responders (improvement)	7 (2,3%)	3 (12%)	3 (3%)	1 (3%)
No Responders	78 (25%)	30 (17%)	37 (37%)	11 (33%)
(stable + worsening)				
Not available	226 (72.7%)	144 (81%)	61 (60%)	21 (64%)
Clinical Disease control 3, *n* (%)				
Disease control	76 (24.4%)	29 (16%)	36 (36%)	11 (33%)
(Stable, improvement)				
No disease control (worsening)	9 (2.9%)	4 (2%)	4 (4%)	1 (3%)
Not available	226 (72.7%)	144 (81%)	61 (61%)	21 (64%)
PET response 4, *n* (%)				
CMR	9 (2.9%)	2 (1%)	4 (45)	3 (9%)
PMR	7 (2.3%)	6 (3%)	1 (1%)	0
SMD	12 (3.9%)	6 (3%)	4 (4%)	2 (6%)
PMD	18 (5.8%)	9 (5%)	6 (6%)	3 (9%)
Not available	265 (85.2%)	154 (87%)	86 (85%)	25 (76%)
Treatment response at PET response 4, *n* (%)				
Responders (CMR, PMR)	16 (5.1%)	8 (4.5%)	5 (5%)	3 (9.1%)
No responders (SMD, PMD)	30 (9.6%)	15 (8.5%)	10 (9.9%)	5 (15.2%)
Not available	265 (85.2%)	154 (87%)	86 (85.1%)	25 (75.8%)
Disease control at PET response 4, *n* (%)				
Disease control	28 (9%)	14 (8%)	9 (9%)	5 (15%)
(CMR, PMR, SMD)				
No disease control (PMD)	18 (5.8%)	9 (5%)	6 (6%)	3 (9%)
Not available	265 (85.2%)	155 (87%)	86 (85%)	25 (76%)
Clinical Response 4, *n* (%)				
Stable disease	33 (10.6%)	15 (8%)	14 (14%)	4 (12%)
Clinical improvement	3 (1%)	1 (1%)	1 (1%)	1 (3%)
Clinical worsening	4 (1.3%)	2 (1%)	2 (2%)	2 (6%)
Not available	271 (87.1%)	159 (90%)	84 (83%)	26 (79%)
Clinical response 4, *n* (%)				
Responders (improvement)	3 (1%)	1 (0.4%)	1 (1%)	1 (3%)
No Responders	37 (11.9%)	17 (9.6%)	16 (16%)	6 (18%)
(stable + worsening)				
Not available	271 (87.1%)	159 (90%)	84 (83%)	26 (79%)
Clinical disease control 4, *n* (%)				
Disease control	36 (11.6%)	16 (9%)	15 (16%)	5 (15%)
(Stable, improvement)				
No disease control (worsening)	4 (1.3%)	2 (1%)	0	2 (6%)
Not available	271 (87.1%)	159 (90%)	84 (83%)	26 (79%)
PET response 5, *n* (%)				
CMR	5 (1.6%)	1 (1%)	2 (2%)	2 (6%)
PMR	1 (0.3%)	1 (1%)	0	0
SMD	8 (2.6%)	5 (3%)	3 (3%)	0
PMD	9 (2.9%)	4 (2%)	2 (2%)	3 (9%)
Not available	288 (92.6%)	166 (94%)	94 (93%)	28 (85%)
Treatment response at PET response 5, *n* (%)				
Responders (CMR, PMR)	6 (1.9%)	2 (1.1%)	2 (2%)	2 (6.1%)
No responders (SMD, PMD)	17 (5.5%)	9 (5.1%)	5 (5%)	3 (9.1%)
Not available	288 (92.6%)	166 (94%)	94 (93%)	28 (84.8%)
Disease control at PET response 5, *n* (%)				
Disease control	14 (4.5%)	7 (4%)	5 (5%)	2 (6%)
(CMR, PMR, SMD)				
No disease control (PMD)	9 (2.9%)	4 (2%)	2 (2%)	3 (9%)
Not available	288 (92.6%)	166 (94%)	94 (93%)	28 (85%)
Clinical Response 5, *n* (%)				
Stable disease	20 (6.4%)	10 (6%)	7 (7%)	3 (9%)
Clinical worsening	2 (0.7%)	0	1 (1%)	1 (3%)
Not available	289 (92.9%)	167 (94%)	93 (92%)	29 (88%)
Clinical response 5, *n* (%)				
Responders (improvement)	0	0	0	0
No Responders	22 (7.1%)	10 (6%)	8 (8%)	4 (12%)
(stable + worsening)				
Not available	289 (92.9%)	167 (94%)	93 (92%)	29 (88%)
Clinical disease control, *n* (%)				
Disease control	20 (6.4%)	10 (6%)	7 (7%)	3 (9%)
(Stable, improvement)				
No disease control (worsening)	2 (0.6%)	0	1 (1%)	1 (3%)
Not available	289 (92.9%)	167 (94%)	93 (92%)	29 (88%)

CMR = complete metabolic response; PMR = partial metabolic response; SMD = stable metabolic disease; PMD = progressive metabolic disease.

## Data Availability

The data presented in this study are available on request from the corresponding author.

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
