# Peer review of "ITA-IMMUNO-PET: The Role of [18F]FDG PET/CT for Assessing Response to Immunotherapy in Patients with Some Solid Tumors"

_cancers, 2023, doi:10.3390/cancers15030878_

Round 1
Reviewer 1 Report
ITA-IMMUNO-PET: The Role of [18F]FDG PET/CT for 2 Assessing Response to Immunotherapy in Patients With Solid 3 Tumors.
This work presented in the manuscript aims to assess immunotherapy responses in patients with solid tumours using (18F)FDG PET/CT.
This is a retrospective study using a relatively high number of patients. Because it is a retrospective design, the authors included the available data, mainly on two types of cancer and a small number of other diseases.
The work and the data provided were produced mainly from two types of solid tumours. The title indicates the techniques used in much broader solid tumours. The title, therefore, should be changed to reflect the work done.
The manuscript contains massive data collected retrospectively by a large number of authors. The study has yet to provide a definite conclusion. I, however, recommend its publication as a step to be followed by further investigations and clinical trials. A complete English revision is to be carried out.
Author Response
ITA-IMMUNO-PET: The Role of [18F]FDG PET/CT for 2 Assessing Response to Immunotherapy in Patients With Solid 3 Tumors.
This work presented in the manuscript aims to assess immunotherapy responses in patients with solid tumours using (18F)FDG PET/CT. This is a retrospective study using a relatively high number of patients. Because it is a retrospective design, the authors included the available data, mainly on two types of cancer and a small number of other diseases.
Q1. The work and the data provided were produced mainly from two types of solid tumours. The title indicates the techniques used in much broader solid tumours. The title, therefore, should be changed to reflect the work done.
R1. The title has been modified as following: ITA-IMMUNO-PET: The Role of [18F]FDG PET/CT for Assessing Response to Immunotherapy in Patients with Some Solid Tumors.
Q2. The manuscript contains massive data collected retrospectively by a large number of authors. The study has yet to provide a definite conclusion. I, however, recommend its publication as a step to be followed by further investigations and clinical trials. A complete English revision is to be carried out.
R2. In the conclusion paragraph, the following sentences have been included: “Prospective randomized controlled trials are needed to establish whether assessing response to therapy early on a combination of [18F]FDG PET/CT and clinical indicators can affect the further management of patients given immunotherapy, and whether it can change their prognosis”. The English revision was made by an expert in the first version. However, after the inclusion of new sentences, some additional revisions have been made.
Reviewer 2 Report
Immunotherapy represents the new arrived in the complex show of oncological treatments. Many questions remain unanswered regarding the best selection of responders and how to evaluate the response. The proposed article has an impressive number of patients treated with this type of medication, evaluated by a very good and modern imagistic technique. However I suggest some clarification :
1- please detail if you observed some differences between primary tumors regarding the uptake of the used tracer - initially and during following examinations (there are some mentions in figure 3 but possibly to improve the content)
2- in the method section, I suggest to explain how you appreciated the response - dimensional or SUV value, both etc. Did you find a cut-off value with statistic significance?
3. I certainly appreciate the subchapter 3.5 but some words regarding the pseudo-progression on immunotherapy in your experience?
4. regarding the interval of evaluations. Due to delay of action for immunotherapy for a period of time between 2 to 6 months, a comparison of median value in the first 2 PET compared with those after 6 months could represent something more appropriate from clinical oncologist point of view?
Author Response
Immunotherapy represents the new arrived in the complex show of oncological treatments. Many questions remain unanswered regarding the best selection of responders and how to evaluate the response. The proposed article has an impressive number of patients treated with this type of medication, evaluated by a very good and modern imagistic technique. However I suggest some clarification :
Q1. Please detail if you observed some differences between primary tumors regarding the uptake of the used tracer - initially and during following examinations (there are some mentions in figure 3 but possibly to improve the content)
R1. The uptake of FDG in the index lesion was calculated in all type of tumors. By comparing the SUVmax in the index lesions we found that the mean (+standard deviation) SUVmax was not statistical different among the pathologies (ANOVA test with Bonferroni analysis), across the first 3 PET controls (baseline PET, PET1 and PET2), being slightly higher in patients with lung cancer, genito-urinary, head and neck, breast, and gastrointestinal cancer than in malignant melanoma. Some small sentences have been added in the text (results part).
Q2. In the method section, I suggest explaining how you appreciated the response - dimensional or SUV value, both etc. Did you find a cut-off value with statistic significance?
R2. The evaluation of response to therapy was made by using the EORTC criteria. We chose to use SUVmax, without using a specific cut-off value, in each PET scan. Therefore, the change in SUVmax, in terms of percentage was computed. Two sentences have been added for explaining it, in the methods section.
Q3. I certainly appreciate the subchapter 3.5 but some words regarding the pseudo-progression on immunotherapy in your experience?
R3. Pseudoprogression has been mainly reported in melanoma patients receiving anti-CLTA4 agents, with approximately 15% of patients experiencing pseudoprogression. Pseudoprogression appears to be much rarer in all other tumour types (less than 3%), especially with the use of anti-PD1/PD-L1 agents, indicating that in most of the patient’s progression seen on morphological imaging is authentic progression. Based on the number reported in Figure 1s, relative to all patients with available data at PET1 and PET2, 11/66 patients demonstrated a potential pseudoprogression at PET1. In particular, 4/11 (36%) patients were affected by lung cancer, while 7/11 (64%) by malignant melanoma. In this latter group of patients, all underwent anti-CTLA4 therapy. The false interpretation of pseudoprogression was relative to the appearance of a slight uptake of FDG in some lymph nodes or in small lung nodules, respectively for melanoma and lung patients. Some new sentences have been included in the results and in discussion paragraphs.
Q4. Regarding the interval of evaluations. Due to delay of action for immunotherapy for a period of time between 2 to 6 months, a comparison of median value in the first 2 PET compared with those after 6 months could represent something more appropriate from clinical oncologist point of view?
R4. This is a good time point that we have included as potential next steps for a clinical trial. For this reason, we have included a small sentence in the conclusion section. In the present study we considered only the EORTC criteria, and not the semiquantitative parameters in terms of SUVmax, MTV, TLG, etcc. We have decided to avoid using the semiquantitative numbers due to the variability among the different scanners but including an objective parameter able to assess the response to immunotherapy. In an ongoing analysis, we are going to collect some semiquantitative variables in terms of ratios between lesions in T,N,M and the SUVmean in some references’ organs (such as liver, mediastinal blood pool and spleen) and their variation between a PET scan and another one after immunotherapy’s start.
Round 2
Reviewer 1 Report
Thank you for completing the corrections required. The paper is now ready for publication.